# Ovarian Morphology in Non-Hirsute, Normo-Androgenic, Eumenorrheic Premenopausal Women from a Multi-Ethnic Unselected Siberian Population

**DOI:** 10.3390/diagnostics14070673

**Published:** 2024-03-22

**Authors:** Ludmila Lazareva, Larisa Suturina, Alina Atalyan, Irina Danusevich, Iana Nadelyaeva, Lilia Belenkaya, Irina Egorova, Kseniia Ievleva, Natalia Babaeva, Daria Lizneva, Richard S. Legro, Ricardo Azziz

**Affiliations:** 1Scientific Center for Family Health and Human Reproduction Problems, 664003 Irkutsk, Russia; 2Icahn School of Medicine at Mount Sinai, New York, NY 10029, USA; 3Penn State College of Medicine, Penn State University, Hershey, PA 17033, USA; 4Heersink School of Medicine, The University of Alabama at Birmingham, Birmingham, AL 35294, USA

**Keywords:** pelvic ultrasound, follicle number per ovary, FNPO, ovarian volume, PCOM, upper normal limits, UNL, healthy population

## Abstract

Polycystic ovary syndrome (PCOS) is a highly prevalent disorder in women, and its diagnosis rests on three principal features: ovulatory/menstrual dysfunction, clinical and/or biochemical hyperandrogenism, and polycystic ovarian morphology (PCOM). Currently, data on age- and ethnicity-dependent features of PCOM remain insufficient. We aimed to estimate ethnicity- and age-dependent differences in ovarian volume (OV) and follicle number per ovary (FNPO) in a healthy, medically unbiased population of Caucasian and Asian premenopausal women, who participated in the cross-sectional Eastern Siberia PCOS epidemiology and phenotype (ESPEP) study (ClinicalTrials.gov ID: NCT05194384) in 2016–2019. The study population consisted of 408 non-hirsute, normo-androgenic, eumenorrheic premenopausal women aged 18–44 years. All participants underwent a uniform evaluation including a review of their medical history and a physical examination, blood sampling, and pelvic ultrasonography. The statistical analysis included non-parametric tests and the estimation of the upper normal limits (UNLs) by 98th percentiles for OV and FNPO. In the total study population, the upper OV percentiles did not differ by ethnicity or age group. By contrast, the UNL of FNPO was higher in Caucasian women than in Asian women, and women aged <35 years demonstrated a higher UNL of FNPO compared to older women. In summary, these data suggest that the estimation of FNPO, but not OV, should take into account the ethnicity and age of the individual in estimating the presence of PCOM.

## 1. Introduction

One of the criteria of polycystic ovarian syndrome (PCOS) in the majority of cases is the polycystic structure of the ovaries [1]. The international guideline for the assessment and management of PCOS patients, published in 2018 [1] and updated in 2023 [2,3], proposed considering the Rotterdam 2003 criteria for PCOM as standard during PCOS diagnosis, and it simultaneously stressed the need to take into account racial and age characteristics.

Previously, ethnic variations in follicle number and/or ovarian volume (OV) have been demonstrated by some authors in different populations. Thus, in Chinese women, as compared to women in the European population, the smaller ovarian volume and the lower number of follicles were proposed as sufficient criteria for diagnosing PCOM: ≥6.3 cm^3^ and ≥10 follicles [3,4]. In Turkish women, the lower ovarian estimates compared to those of the Western female population were found with the following threshold criteria for PCOM: an ovarian volume of 6.43 cm^3^ and a number of follicles ≥ 8 [5]. According to the current guideline for the assessment and management of PCOS patients, for PCOM diagnosis, the ovarian volume is a much more reliable indicator than the number of follicles [1,2]. Nevertheless, in the population of Korean women, the number of follicles was considered to be a more significant criterion for polycystic disease than the volume of the ovary, due to the smaller volume of the ovaries in the Asian race [6,7].

Evidently, age-related processes in women suggest a reduction in the number of growing antral follicles. The volume of the ovaries and the number of follicles depend on the lifespan of the reproductive period, reaching a maximum value in adolescence, with a gradual decrease in adulthood and a fast decrease at the age of menopause [8,9,10,11,12]. At the same time, the number of follicles decreases faster than OV. Regarding PCOM, in women aged ≥35 years, the prevalence of polycystic ovaries is 7.8% vs. 21% in younger women [7,8].

According to the international guidelines on the diagnosis and management of patients with PCOS, transvaginal ultrasonography should be performed in the early follicular phase of the natural cycle or after withdrawal bleeding caused by pharmaceuticals. Currently, the criteria for PCOM in women aged 18–35 years are as follows: ≥20 follicles in at least one ovary and/or ovarian volume ≥ 10 mL, without the presence of dominant follicles, cysts, or corpus luteum [1,2,13]; this approach to follicle number estimation is applicable if a transducer above 8 MHz is used. However, in clinical practice, as well as in epidemiological studies, equipment with a sensor frequency of 4–8 MHz is still widely used [1,2,13].

As previously shown in epidemiological studies, diagnostic criteria for PCOM based on ovarian volume (OV) and follicle number per ovary (FNPO) could be determined using different approaches: (a) by performing receiver operator characteristic (ROC) curve analyses (which report the diagnostic power of a parameter to distinguish between diseased and non-diseased conditions and propose thresholds that balance test sensitivity and test specificity) [14] or (b) by means of cluster analysis in a large population-based unselected cohort. Some authors also utilized the upper (95th–98th) percentiles in a well-characterized cohort of women with regular predictable menstrual cycles of 21–35 days in length and no clinical and/or biochemical evidence of hyperandrogenism (HA), recruited from the same population and examined in a similar manner as the study subjects [14] to establish diagnostic criteria for polycystic ovarian morphology.

In general, the data on age- and ethnicity-dependent diagnostic criteria for PCOM remain insufficient and may vary in different geographical zones. In this study, we aimed to estimate the upper percentiles for OV and FNPO in an unselected sub-population of healthy, non-hirsute, normo-androgenic, eumenorrheic premenopausal Eastern Siberian women to determine the need for ethnicity- and age-dependent diagnostic criteria for PCOM.

## 2. Materials and Methods

Study design and population. Study subjects were recruited during the cross-sectional institution-based prospective Eastern Siberia PCOS epidemiology and phenotype (ESPEP) study (ClinicalTrials.gov ID: NCT05194384), conducted in two major areas of Eastern Siberia (the Irkutsk region and the Buryat Republic, Russian Federation) from March 2016 to December 2019, as previously described [15,16]. In brief, a total population of 1490 premenopausal women underwent a mandatory annual employment health assessment. Of these, all women aged 18–44 years who were found to be completely healthy constituted the study population (Figure 1). The inclusion criteria for the current study population included a history of regular, predictable menstrual cycles of 21–35 days in length and no clinical signs of HA and/or elevated testosterone and/or DHAS levels and/or FAI [16]. Women with a BMI < 18 or ≥30 kg/m^2^, premature ovarian failure (based on history or due to elevated FSH), treated and untreated hyperprolactinemia (based on history or increased prolactin level > 727 mIU/mL), untreated thyroid disorder (based on history or TSH level > 4 mIU/mL), and 21-hydroxylase deficient non-classical adrenal hyperplasia (based on increased 17-hydroxyprogesterone (17-OHP) > 7.0 nmol/L), were excluded. The main characteristics of the study population, including their socio-demography, menstrual and reproductive history, anthropometry, vital signs, and hormone profiles, in total and by ethnicity, are presented in the Appendix A.

*Study Protocol*. As previously described, subjects were evaluated consecutively by trained personnel and by means of questionnaires, anthropometry, vital signs, and gynecological examination. Anthropometry measurements included height, weight, and waist circumference (WC). Their body mass index (BMI) was calculated as weight (kg)/height (m^2^). Hirsutism was defined by the modified Ferriman–Gallwey (mFG) visual hirsutism score scale [15,17].

Hormonal analyses. Blood samples were obtained in the morning, and analyzed for serum total testosterone (TT), DHEAS, sex hormone-binding globulin (SHBG), prolactin, TSH, LH, FSH, and 17-OHP, and the free androgen index (FAI) was calculated (i.e., [TT/SHBG] × 100).

We used a validated, highly efficient liquid chromatography–tandem mass spectrometry (LC-MS/MS) assay (Shimadzu, Kioto, Japan) in positive polarity mode using a dual ionization source (DUIS) to determine TT. The chromatography was performed with a Kromasil 100-2.5-C18 column (2.1 mm × 100 mm, AkzoNobel, Bohus, Sweden) and an isocratic elution mode using a mobile phase consisting of acetonitrile and 0.1% aqueous solution of formic acid. [2H3]-testosterone (ALSAchim, Strasbourg, France) was used as the internal standard for the assay. The lower limit of TT quantification was 5 ng/dl with an average accuracy of 100.23%. Serum levels of SHBG, prolactin, FSH, LH, TSH, and 17-OHP were assessed by enzyme-linked immunosorbent assay (ELISA) (ELx808, Bio-Tek Instruments, Winooski, VT, USA), using kits manufactured by AlcorBio (Russia). Serum DHEAS was detected using a competitive chemiluminescent enzyme immunoassay (Immulite 1000, Siemens, Randolf, Utah, USA). Anti-Müllerian hormone (AMH) was assessed by ELISA using kits produced by Xema Co., Ltd. (Moscow, Russia).

*Pelvic ultrasound (U/S)* was performed by experienced specialists who were trained to conduct the U/S scans uniformly, with the appropriate inter-observer variability. We used Mindray M7 (Mindray Bio-Medical Electronics Co., Shenzhen, China), a transvaginal probe (5.0–8.0 MHz). Ovarian volume was determined by the formula for a prolate ellipsoid (length × width × height × 0.523) [18].

*Statistical analysis.* As previously reported, sample size calculations for the total population of the ESPEP study were based on the following formula: $n={\left(z_{1-\alpha }\right)}^{2}(P\left(1-P\right))/D^{2}$, where n = individual sample size, z_(1 − α) = 1.96 (when α = 0.05), P = assumed PCOM prevalence for unselected population according to previously published data, D = error. If we take prevalence as 33% [8,9] (or 0.33) and absolute error as 5%, then the minimum sample size is as follows:$$n=\frac{1.96^{2\;}\left(0.33\times \left(1-0.33\right)\right)}{0.05^{2}}\approx 340$$

Data were collected using research electronic data capture (REDCap) [19].

Managing missing data: In our research dataset, there were two types of missing data: missing completely at random (MCAR) and missing at random (MAR). We recorded all missing values with labels of “N/A” to make them consistent throughout our dataset. When analyzing the dataset, we used the pairwise deletion method.

The results of the Kolmogorov–Smirnov test for normality showed that the analyzed continuous variables were non-normally distributed. Therefore, for continuous variables, we used Mann–Whitney non-parametric tests. Kruskal–Wallis ANOVA and z-criteria were used to compare proportions and categorical variables. A *p*-value of 0.05 was considered statistically significant. To compare the 95th, 97.5th, and 98th percentiles, we analyzed 95% confidential intervals (95% CIs). For the construction of 95% CIs we utilized the bootstrap percentile method. Overlapping 95% CIs can explain statistical significance when comparing two measured results. If the two 95% CIs do not overlap, we considered 95th–98th percentiles significantly different [14].

## 3. Results

After exclusions, 444 healthy women (285 Caucasians, 123 Asians, and 36 of mixed ethnicity) were eligible to be included in the study population. Taking into account the low number of women of mixed ethnicity, we further excluded these from the study, leaving 408 women for analysis (Figure 1).

The mean age of the study population with regular predictable menstrual cycles and no clinical or biochemical evidence of hyperandrogenism was 34.32 ± 5.96 years. Women of Caucasian and Asian ethnicity were comparable by age, anthropometric characteristics, and marital status, although these groups exhibited some differences in respect of education and occupation. Participants of Asian origin demonstrated a lower mFG score, within a normal range. Regarding serum FSH, LH, TSH, 17OHP, and AMH levels, no statistically significant differences were detected. At the same time, prolactin levels were significantly higher (within normative ranges) in Asians as compared to Caucasian women. When studying the impact of ethnicity on androgens, we found that TT, DHAS, and FAI values in the study population were significantly lower in Asians than in Caucasian women, but they were in the normative ranges for the multi-ethnic Siberian population [16]. At the same time, Asians showed lower levels of SHBG (Appendix A).

In all the study participants we analyzed, OV and FNPO for ovaries satisfied the following criteria: (a) an absence of follicles and/or cysts greater than 9 mm in diameter, (b) an absence of corpus luteum, and (c) an absence of ovarian masses. Finally, among the total number of healthy premenopausal women from the unselected population (*n* = 408), the data for 563 ovaries were eligible for further investigation. For these ovaries, we performed a descriptive statistical analysis and determined the 98th percentiles for OV and FNPO.

All estimates of OV and FNPO are shown in Table 1 and Table 2 in totals, by ethnicity and age. As presented in the tables, the upper percentiles of both OV and FNPO were calculated for the total group and for subgroups of Caucasian and Asian ethnicity < 35 and ≥35 years old. Based on the calculation of 95% CIs for these percentiles and on the analysis of their overlapping, we compared UNLs (the 98th percentiles).

In the total study population, the upper OV percentiles did not differ by ethnicity or age group; by contrast, the UNL of FNPO was higher in Caucasian women than in Asian women (Table 1 and Table 2). At the same time, women from the study sub-population aged <35 years demonstrated a higher UNL of FNPO compared to older women; this was observed mainly in the Caucasian women. In the Caucasian group, we found the higher 98th percentile for FNPO in women younger than 35 years, whereas the upper percentiles calculated for FNPO in women of Asian ethnicity did not vary by age (Table 2).

## 4. Discussion

Ovarian morphology (specifically, OV and FNPO) is one of the key characteristics of polycystic ovaries; however, ultrasound features of PCOM are not rare and are observed in up to 16–25% of healthy women with regular menstrual cycles [8]. As previously reported, the proposed thresholds for FNPO and OV were not similar for populations of different ethnicity (19–30) (Table 3). Nevertheless, establishing the ethnicity-specific diagnostic criteria of polycystic ovarian morphology is still challenging. In our study, we based our research on determining the upper (98th) percentile of both OV and FNPO in non-hirsute, normo-androgenic, eumenorrheic premenopausal women from a multi-ethnic unselected Siberian population.

Our data demonstrated that, in terms of means, OV was increased in Caucasian women as compared to Asians (6.58 ± 2.36 vs. 5.69 ± 2.09); however, for upper percentiles, the difference was not statistically significant, due to the overlap of 95% CIs. For OV, the upper percentiles determined for Caucasians in our study were comparable to the same estimate in the population-based study conducted by Lujan et al. (2013) [30] in the United States and Canada. At the same time, according to the previously reported data based on ROC analysis, UNLs for OV in French Caucasians [25], Indian women [26], women from the United States and Iceland [27], and Turkish and Vietnamese women [5,28] were lower than our estimates. However, most of these studies were performed in the relatively small hospital-based samples (Table 3).

Regarding FNPO, we demonstrated a substantially higher threshold of FNPO at upper percentiles in women of Caucasian origin as compared to Asians. In our study, the UNL of FNPO for Caucasians was slightly higher (15 vs. 12) as compared to that which had previously been demonstrated [23,25,26,29], and was consistent with UNLs reported by Ahmad et al. (2019, USA) [20]. At the same time, Köşüş et al. (2011) [5] presented significantly lower values for FNPO for the control group of Caucasians as compared to our data.

Our estimates of FNPO for the Asian reference group were consistent with data previously published by Chen et al. (2008) [22], who used ultrasound equipment of a similar class as that used in our study. Alternatively, our data, as well as findings from the Chinese study mentioned above, differ from those reported for a Thai Asian population by Wongwananuruk et al. (2018) [31].

The changes in the follicle number and OV with age were previously observed in the general populations [20,32,33,34]. Moreover, the development of age-specific diagnostic criteria for PCOM in women of reproductive age had been recommended by the international guidelines [1,2]. Nevertheless, the data regarding age-dependent thresholds for ovarian morphology were insufficient [20,27,29]. In our study, in the total population of healthy women, the upper percentiles for FNPO in women aged <35 years and ≥35 years differ significantly (15 vs. 12), mainly due to the Caucasians (15 vs. 13). At the same time, we have not found age-dependent differences in FNPO in the Asian subgroup (11 vs. 10, respectively).

*Study strengths.* Importantly, our study benefited from the fact that we identified study subjects with regular, predictable menstrual cycles and no clinical or biochemical evidence of hyperandrogenism (reference group) in a representative unselected, medically unbiased, multi-ethnic population of women, living in the same geographical and socioeconomic conditions. We consider the Eastern Siberian population as an appropriate model for the purpose of studying the ethnicity-dependent aspects of ovarian morphology in Caucasians and Asians. All study participants were well phenotyped, with the exclusion of any factors that could influence their PCOM characteristics. In addition, all measurements (FNPO, OV) were accomplished by only three specialists who were trained to conduct the U/S scans uniformly with the same machines to eliminate potential bias. Therefore, the proposed criteria could be useful and convenient for diagnosing PCOM. *Study limitations*. While large, our study cohort is still relatively limited in size, and the results should be confirmed in much larger populations. The use of the US equipment with probes of ≤8 MHz is one of our study limitations as well. At the same time, mid-range equipment is most commonly utilized in standard clinical practice, and we suggest that our data on OV and FNPO thresholds are still valid, although the use of probes of >8 MHz is highly recommended [2]. We were also not able to estimate ultrasound characteristics in women of mixed Caucasian/Asian ethnicity, due to the insufficient number of participants.

## 5. Conclusions

In this study, we found that, in the total study population of healthy women, OV based on the upper percentiles did not depend on ethnicity, whereas estimates of FNPO were significantly higher in Caucasians as compared to Asians. We did not find age-dependent differences for OV estimates in the total study population. By contrast, for FNPO, we demonstrated higher upper percentiles in women aged <35 vs. women ≥35 years old. Our data suggest that the estimation of PCOM should take into account the ethnicity and age of the individual.

## Figures and Tables

**Figure 1 diagnostics-14-00673-f001:**
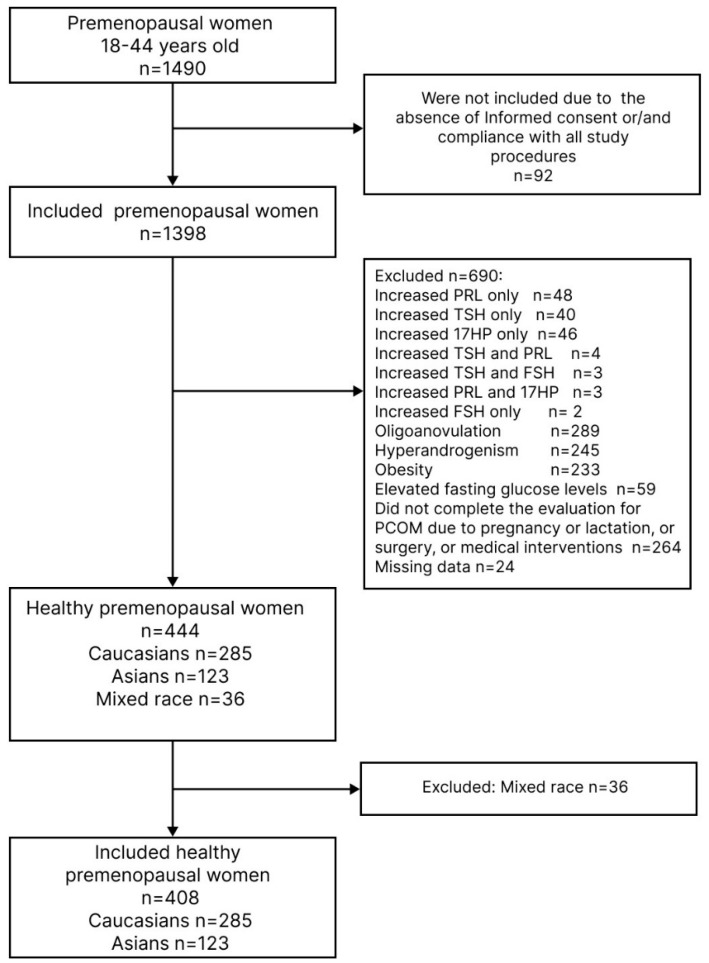
Flow diagram of selection of healthy premenopausal women among the participants in the cross-sectional Eastern Siberia PCOS epidemiology and phenotype (ESPEP) study.

**Table 1 diagnostics-14-00673-t001:** Ovarian volume (OV) and follicle number (FNPO) in healthy women from unselected population.

	Total *N* = 408	Caucasians *N* = 285	Asians *N* = 123	*p*-Value
	*n* = 563 *	*n* = 388 *	*n* = 175 *	
OV				
Mean ± SD	6.30 ± 2.31	6.58 ± 2.36	5.69 ± 2.09	** *p_U_ < 0.001* **
(Min–Max)	(0.54; 16.98)	(0.54; 16.98)	(1.57; 14.63)	
Median	6.01	6.305	6.00	
(Lower Q; Upper Q)	(4.77; 7.37)	(5.04; 7.78)	(4.34; 6.63)	
95 Percentile (95% CI)	10.31 (9.86; 11.22)	10.63 (10.01; 11.88)	9.32 (8.57; 10.65)	
97.5 Percentile (95% CI)	12.3 (10.68; 13.16)	12.45 (11.09; 13.30)	10.62 (9.34; 13.92)	
98 Percentile (95% CI)	12.56 (11.28; 13.56)	12.58 (11.39; 13.76)	10.66 (9.51; 14.29)	
FNPO				
Mean ± SD	6.85 ± 2.78	7.19 ± 3.00	6.11 ± 2.03	** *p_U_ < 0.001* **
(Min–Max)	(1.00; 30.00)	(1.00; 30.00)	(1.00; 14.00)	
Median	6.00	7.00	6.00	
(Lower Q; Upper Q)	(5.00; 8.00)	(5.00; 8.00)	(5.00; 6.00)	
95 Percentile (95% CI)	12 (10.00; 10.72)	** *12 (10; 12)* **	** *10 (9; 10) #* **	
97.5 Percentile (95% CI)	14 (12.00; 14.95)	** *14 (12; 14)* **	** *10 (9; 10) #* **	
98 Percentile (95% CI)	14 (12.00; 14.00)	** *15 (13.25; 15.26)* **	** *10.52 (10; 12) #* **	

* n—number of ovaries available for evaluation; U—Mann–Whitney non-parametric test; #—significant difference between groups, based on the estimation of overlapping of 95% CIs.

**Table 2 diagnostics-14-00673-t002:** Ovarian volume (OV) and follicle number (FNPO) in healthy women from unselected population by age.

	Total *N* = 408		Caucasians *N* = 285		Asians *N* = 123	
	<35 yrs *n* = 269 *	≥35 yrs *n* = 294 *	<35 yrs *n* = 194 *	≥35 yrs *n* = 194 *	<35 yrs *n* = 75 *	≥35 yrs *n* = 100 *
Ovarian volume						
Mean ± SD	6.72 ± 2.37	5.91 ± 2.2 **	7.09 ± 2.38	6.07 ± 2.22 **	5.78 ± 2.07	5.62 ± 2.12
(Min–Max)	(0.54; 16.98)	(0.94; 14.63)	(0.54; 16.98)	(0.94; 13.56)	(1.57; 12.72)	(2.2; 14.63)
Median	6.28	5.62	6.7	5.93	5.22	5.4
(Lower Q; Upper Q)	(5.08; 7.85)	(4.39; 7.11)	(5.5; 8.09)	(4.6; 7.27)	(4.44; 6.96)	(4.17; 6.46)
95 Percentile	11.32	9.83	12.05	9.85	9.55	9.18
(95% CI)	(10.13; 12.65)	(9.32; 10.32)	(10.53; 12.89)	(9.49; 10.26)	(8.43; 11.26)	(7.96; 10.88)
97.5 Percentile	12.67	10.47	12.81	10.24	10.2	10.64
(95% CI)	(11.38; 13.81)	(9.85; 12.56)	(11.6; 14.18)	(9.85; 12.56)	(9.11; 12.72)	(8.59;14.63)
98 Percentile	12.73	10.6	13.26	10.35	10.39	10.74
(95% CI)	(11.91; 15.15)	(9.94; 13.56)	(12.33; 15.9)	(9.87; 12.56)	(9.22; 12.72)	(9.02; 14.63)
Follicle number per ovary (FNPO)						
Mean ± SD	7.88 ± 2.9	5.91 ± 2.29 **	8.22 ± 3.14	6.15 ± 2.47 **	7.03 ± 1.94	5.43 ± 1.83 **
(Min–Max)	(3; 30)	(1; 15)	(3; 30)	(1; 15)	(3; 14)	(1; 12)
Median	7	6	7	6	7	5
(Lower Q; Upper Q)	(6; 9)	(5; 7)	(6; 9)	(5; 7)	(6; 8)	(4; 6)
95 Percentile	13	10.35		12	10	8.05
(95% CI)	(12.0;14.6)	(9.0; 12.0)	14 (13.0; 16.0)	(11.65; 14.65)	(8.0; 11.0)	(8.0; 10.05)
97.5 Percentile	15	12	15	12.17	10.3	9.52
(95% CI)	(13.0; 16.0)	(11.0; 13.32)	(12.83; 16.0)	(12.0; 14.0)	(10.0; 14.0)	(8.0; 11.52)
98 Percentile	* **15** *	* **12 #** *	* **15.4** *	* **13 #** *	11.04	10.02
(95% CI)	* **(13.0; 16.0)** *	* **(10.0; 12.0)** *	***(14.0; 18.0**)*	*(**12.0; 14.0)***	(10.0; 14.0)	(8.02; 12.0)

*—number of ovaries available for evaluation; **—pu < 0.001; U—Mann–Whitney non-parametric test; #—significant difference between groups <35 and ≥35 years, based on the estimation of overlapping of 95% CIs.

**Table 3 diagnostics-14-00673-t003:** Thresholds for follicle number and ovarian volume proposed by different authors.

Author, Year, Country	Setting Study Design #	Total Population	Ethnicity Controls	Age Range	OV, Mean ± StD. (Min–Max) For Controls	OV, UNLs Controls	FNPO Mean ± SD. (Min–Max) For Controls	FPNO, UNLs Controls	Transducer Frequency
Ahmad et al., 2019, USA [20].	Cross-sectional study	Control: 756 (FNPO, OV) PCOS: 245 (FNPO), 297 (OV)	Caucasians	Overall (20–40)	6.49 ± 4.98	6.75	10.01 ± 5.29	13	4–8 MHz 4–10 MHz
				25 to <30	7.31 ± 6.33	8.5	12.38 ± 5.52	15	
				30 to <35	6.49 ± 4.97	7.00	10.14 ± 4.8	14	
				35 to <40	5.82 ± 3.39	6.25	7.96 ± 4.66	12	
Carmina et al., 2016, Italy [21].	Retrospective matched controlled study	PCOS: 113 Control: 47	Caucasians	19 to 35 years	N/A	4.4 ± 1.8	N/A	10 ± 4	8–10 MHz
Chen et al., 2008, China [22].	Age-matched women	PCOS: 432 Control: 153	Chinese population	N/A	N/A	6.4	N/A	10	6 MHz
Dewailly et al., 2014, France [23].	Retrospective study	Control: 521 PCOS: 272 OA + HA (full-blown): 95 OA + PCOM: 110 HA + PCOM: 67	Caucasians	18 to 40 years	N/A	N/A	N/A	12.0	5–7 MHz
Fulghesu et al., 2001, Italy [24].	Retrospective data analysis	Control: 30 Multi-follicular Ovaries (MFO): 27 PCOS: 53	Caucasians	18–38	N/A	13.21	N/A	N/A	6.5 MHz
Jonard et al., 2005, France [25].	Observational cohort study	Control: 57 PCOS: 98	Caucasians	Control: 29.0 (24.5–35.0) PCOS: 27.2 (19.5–33.0)	4.75 (3.11–6.86)	7	6.5 (4.5–10.5)	12.0	7 MHz
Sujata and Swoyam, 2018, India [26].	Not provided	PCOS: 86 Control: 45	Caucasians	18–45 years	5.06 ± 2.44	6.15	7.13 ± 3.51	12.0	6–12 MHz
Kim et al., 2017, United States/Iceland [27].	Cross-sectional, case-control design	Control: 666 (Boston) and 32 (Iceland) PCOS: 544 (Boston) and 105 (Iceland) 18 to >44 years	Caucasians	≤24 years	N/A	12	N/A	13	4–8 MHz
				25–29 years	N/A	10	N/A	14	
				30–34 years	N/A	9	N/A	10	
				35–39 years	N/A	8	N/A	10	
				40–44 years	N/A	10	N/A	9	
Köşüş et al., 2011, Turkey [5].	Prospective study	Control: 65 PCOS: 251	Caucasians		N/A	6.43	N/A	8	6.5 MHz
Le et al., 2021, Vietnam [28].	Cross-sectional study	Control: 273 PCOS: 119	Asians	33.99 ± 4.78 years	6.08 ± 3.67	6.0	N/A	N/A	7 MHz
Lie Fong et al., 2017, Netherlands/United States [29].	Retrospective observational cohort study	Control: 297 Young non-PCOM (Cluster 1): 118 Young PCOM (Cluster 2): 28 Old non-PCOM (Cluster 3): 100 Old PCOM (Cluster 4): 51 PCOS: 700	Caucasians	Young women	N/A	N/A	9 (5–24)	12.25	6.5–8 MHz
				Old women	N/A	N/A		10.75	
Lujan et al. 2013, United States/Canada [30].	A diagnostic test study was performed using cross-sectional data	Control: 70 PCOS: 98	Caucasians	18–44 years	N/A	10	N/A	26	5–9 MHz 6–12 MHz
Wongwananuruk et al., 2018, Thailand [31].	Cross-sectional study	Control: 63 PCOS: 55	Asians	18–45 years of age	4.66 ± 1.83	6.5	9.97 ± 3.86	15	8 MHz

N/A—Not available.

## Data Availability

Data from the study may be available to other researchers who have developed important research questions that can be answered by these valuable data. This data access policy applies to all individuals or organizations who would like to utilize data from the study. Data may be requested by researchers from various institutions for research purposes only by submitting an expression of interest (EoI), which should include brief information about a project leader’s name, the institution, a title of the potential project, ethical approval from the ethics committee, and a summary of the proposed project. Individual participant data to be shared may include de-identified socio-demographic and clinical data, as well as lab tests results.

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
