# Peer review of "Ovarian Morphology in Non-Hirsute, Normo-Androgenic, Eumenorrheic Premenopausal Women from a Multi-Ethnic Unselected Siberian Population"

_diagnostics, 2024, doi:10.3390/diagnostics14070673_

Round 1

Reviewer 1 Report

Comments and Suggestions for Authors

Dear authors,

I have read the article title “Ovarian morphology in non-hirsute, normo-androgenic, eumenorrheic premenopausal women from a multi-ethnic unselected Siberian population” with great interest. The idea behind the work is really important and was the need for the diagnosis of PCOM. The authors have documented their work in very professional way and there is little rather no English or grammar mistakes. The flow of the article was maintained thoroughly throughout the manuscript. However, I have some comments which need to be addressed.

In the abstract define PCOSM and FNPO at first come. Similarly, OV has not defined previously.

In the method section the authors have calculated the CV for ultrasonographic findings. However, no CV has given for hormone analysis the used for analysis of different hormones.

The use of tenses needs to be uniform. The authors sometime represent the results in present tense and sometimes in past like” At the same time, prolactin levels are significantly higher (within normative ranges) in Asians as compared to Caucasians women”.

In my opinion the representation of 1 and 2 for subgroup is misleading. Especially in the calculation of p value. The authors are calculating the difference between Caucasian and Asians which is well understood. Therefore, such representation sometimes led to confusion in the data interpretation.

In the tables the authors describe the p value in some cases as p1-2=0.000*. In my opinion a lowest p value such as given should be described as p<0.001.

 I found no text description for table 4. The representation of Table 4 is not relevant to the present work described by authors though it is of much importance. This Table need to be shifted in to supplementary materials or need clear in text description how this table add to the present study.

In the discussion section the authors write “In general, the incidence of PСOM in the unselected populations is 33-22% (8-10).’ This statement is rather vague and need clarity.

Author Response

The authors thank the reviewer for very helpful suggestions. We have attempted to respond and incorporate to all suggestions. A point-by-point response is below: 

In the abstract define PCOSM and FNPO at first come. Similarly, OV has not defined previously.

-When abbreviations are used for the first time, the full versions of the terms have been indicated.

In the method section the authors have calculated the CV for ultrasonographic findings. However, no CV has given for hormone analysis the used for analysis of different hormones.

- Indeed, the coefficients were calculated for hormones and reported separeately, regarding U/S variance assessed the correct term is “inter-observer variability”.

The use of tenses needs to be uniform. The authors sometime represent the results in present tense and sometimes in past like” At the same time, prolactin levels are significantly higher (within normative ranges) in Asians as compared to Caucasians women”.

-Thank you. We have corrected all tenses as past.

In my opinion the representation of 1 and 2 for subgroup is misleading. Especially in the calculation of p value. The authors are calculating the difference between Caucasian and Asians which is well understood. Therefore, such representation sometimes led to confusion in the data interpretation.

-We have edited the abstract and the body of the manuscript to clarify this issue.

In the tables the authors describe the p value in some cases as p1-2=0.000*. In my opinion a lowest p value such as given should be described as p<0.001.

-Agree. Done

I found no text description for table 4. The representation of Table 4 is not relevant to the present work described by authors though it is of much importance. This Table need to be shifted in to supplementary materials or need clear in text description how this table add to the present study.

-We have moved previous Table 4 to Supplemental data.

In the discussion section the authors write “In general, the incidence of PСOM in the unselected populations is 33-22% (8-10).’ This statement is rather vague and need clarity.

-This typo was corrected

Reviewer 2 Report

Comments and Suggestions for Authors

The authors have undertaken to estimate the ovarian volume and FNPO in an unselected population of premenopausal Eastern Siberian women to justify the need to determine the ethnicity and age-dependent diagnostic criteria for PCOM. Based on the results presented, the authors suggest that in multiethnic, Caucasian/Asian populations, the age- and ethnicity issues regarding FNPO should be taken into account when determining the diagnostic PCOM criteria. The age-dependent difference is also important for the estimation of OV in Caucasians. The structure of the paper is correct. The authors well described the objectives, and the methods used and presented the results obtained. The discussion describes the obtained results against the background of the available literature. The statistical analyses are based on the data in several tables. The authors statistically compare some groups with each other as shown in the results chapter. However, it is hard for the reader looking through these tables to detect these differences. Therefore, I would suggest highlighting these differences in the tables more by enlarging the symbols with which these differences are marked or changing the color of these symbols. The authors also show the p= values in these tables. Some of the values are marked p 1-2 while others are marked p 1a-1b. Is it a matter of numbering the groups being compared? This would need to be described in more detail in the table description. Also, some of the p-values are zero, or is the p-value so small that the number of positions after the dot is inadequate to present the p-value? Consideration of these changes will significantly improve the readability of the publication which will allow it to be published in Diagnostics.

Author Response

The authors statistically compare some groups with each other as shown in the results chapter. However, it is hard for the reader looking through these tables to detect these differences. Therefore, I would suggest highlighting these differences in the tables more by enlarging the symbols with which these differences are marked or changing the color of these symbols. The authors also show the p= values in these tables. Some of the values are marked p 1-2 while others are marked p 1a-1b. Is it a matter of numbering the groups being compared? This would need to be described in more detail in the table description. Also, some of the p-values are zero, or is the p-value so small that the number of positions after the dot is inadequate to present the p-value? Consideration of these changes will significantly improve the readability of the publication which will allow it to be published in Diagnostics.

The authors thank the reviewer for very helpful suggestions. We have attempted to incorporate to all suggestions. We have modified the table formats to make it easier for the reader to detect when differences exist.

Reviewer 3 Report

Comments and Suggestions for Authors

As the PCOS is important, challenging and cost-consuming medical issue ongoing studies are of high importance and scientifically expected. Presented study seems to be properly designed and well conducted. As general concern of mine is population usefulness of the obtained results, as ethnic differences in terms of morphological, biochemical and functional parameters are to be expected. The results should be treated, considering the size of the group and predictions (research hypotheses), as an introduction to a much larger project, although over 400 patients is a large group, even for PCOS.

Maybe it would be good idea to mention this, because unified thresholds in many cases are problematic…

I would recommend study for publication after introduction so amendments into the manuscript. My remarks are enlisted below.  

- abbreviations as well as acronyms in their first use should be given In full names.

- p-values <0.05 are commonly treated as statistically significant levels so e.g. p=0.311 cannot be counted as significant

- in the table “p1-2 “ is useless as obviously comparison was made between pointed two groups.

- moreover comparison between groups/features like occupation or education is useless in regard to title of the study and proposed aim.

- table 2 is composed mostly by crucial results, rest results may be presented but the Authors should revise/analyze what should/may be compared… In regard to the aim of the study mostly…

- inter/intra observer coefficients were less than 6%, is there any standard of examination for such situation if the Authors pointed this out?

- often occurring mistake, e.g. l.172-175 – this are not results, this is study group characteristic and should be properly placed in the M&M section.

- I would negotiate about no. of specimens included into the study… The Authors screened 1490/1398 suspects but enrolled so included into the study 408 patients and this number should be used as study specimens quantity…

- l.97 paragraph and l.110 paragraph are sensu stricto the same characteristics of the study group so they compose inclusion/exclusion criteria in one paragraph.

- usage of dots instead of coma in numerical scale.

I would recommend acceptance for publication after introduction amendments/minor revision.

Author Response

-The authors thank the reviewer for the positive estimate of our research and helpful suggestions. A point-by-point response  is as follows:

As general concern of mine is population usefulness of the obtained results, as ethnic differences in terms of morphological, biochemical and functional parameters are to be expected. The results should be treated, considering the size of the group and predictions (research hypotheses), as an introduction to a much larger project, although over 400 patients is a large group, even for PCOS. Maybe it would be good idea to mention this, because unified thresholds in many cases are problematic…Overall, I am concerned about the population utility of the findings as ethnic differences in terms of morphological, biochemical and functional parameters are to be expected. The results should be considered, given the group size and predictions (research hypotheses), as an introduction to a much larger project, although over 400 patients is a large group, even with PCOS. It might be a good idea to mention this, because uniform thresholds are problematic in many cases.

- We appreciate the reviewer’s comments. We now mention the fact that, while large, our study cohort is still relatively limited in size and the results should be confirmed in much larger populations.

I would recommend study for publication after introduction so amendments into the manuscript. My remarks are enlisted below.  

- abbreviations as well as acronyms in their first use should be given In full names.

-When abbreviations are used for the first time, we now make sure that the abbreviation is explained.

- p-values <0.05 are commonly treated as statistically significant levels so e.g. p=0.311 cannot be counted as significant

-The typo was corrected.

- in the table “p1-2 “ is useless as obviously comparison was made between pointed two groups.

- Corrected.

- moreover comparison between groups/features like occupation or education is useless in regard to title of the study and proposed aim. - table 2 is composed mostly by crucial results, rest results may be presented but the Authors should revise/analyze what should/may be compared… In regard to the aim of the study mostly…

- We moved table 4 to Supplement and presented the comparisons from tables 1-2 (previously: tables 2-3) in the text

- inter/intra observer coefficients were less than 6%, is there any standard of examination for such situation if the Authors pointed this out?

- In reviewing the literature, other investigators used kappa-Statistics, Lin's concordance correlation coefficients or Pearson coefficient and Bland-Altman graphs for the total study populations (Amer SA, et al. An evaluation of the inter-observer and intra-observer variability of the ultrasound diagnosis of polycystic ovaries. Hum Reprod. 2002 Jun;17(6):1616-22.; Lujan, M.E., et al. Assessment of ultrasonographic features of polycystic ovaries is associated with modest levels of inter-observer agreement. J Ovarian Res 2009 2, 6; Kilani R., et al.  Inter-observer variability in the assessment of ultrasound features of polycystic ovaries.  Middle East Fertility Society Journal 22 (2017) 226–232).  Alternatively, in our setting, in addition to having experienced ultrasonographers perform the U/S, all three assessed a small cohort of five subjects, giving us the estimate of variance, based on the average percentage of deviation measurements between specialists.

- often occurring mistake, e.g. l.172-175 – this are not results, this is study group characteristic and should be properly placed in the M&M section.

- Have corrected. We have moved the phrase “The main characteristics of the study population, including their socio-demography, menstrual and reproductive history, anthropometry, vital signs, and hormone profiles, in total and by ethnicity, are presented in the Supplement (Table S1)” to M&M section

  - I would negotiate about no. of specimens included into the study… The Authors screened 1490/1398 suspects but enrolled so included into the study 408 patients and this number should be used as study specimens quantity…

-We have edited the text as suggested. Currently, we present 408 women as a “study population”

 - l.97 paragraph and l.110 paragraph are sensu stricto the same characteristics of the study group so they compose inclusion/exclusion criteria in one paragraph.

-We have edited this section for simplicity and clarity.

 - usage of dots instead of coma in numerical scale.

-Done